# The Innate Immune System in Cardiovascular Diseases and Its Role in Doxorubicin-Induced Cardiotoxicity

**DOI:** 10.3390/ijms232314649

**Published:** 2022-11-24

**Authors:** Anchit Bhagat, Pradeep Shrestha, Eugenie S. Kleinerman

**Affiliations:** Department of Pediatrics-Research, The University of Texas MD Anderson Cancer Center, Houston, TX 77054, USA

**Keywords:** doxorubicin, doxorubicin-induced cardiotoxicity, neutrophils, neutrophil elastase, innate immunity, cardiovascular disease, anthracyclines

## Abstract

Innate immune cells are the early responders to infection and tissue damage. They play a critical role in the initiation and resolution of inflammation in response to insult as well as tissue repair. Following ischemic or non-ischemic cardiac injury, a strong inflammatory response plays a critical role in the removal of cell debris and tissue remodeling. However, persistent inflammation could be detrimental to the heart. Studies suggest that cardiac inflammation and tissue repair needs to be tightly regulated such that the timely resolution of the inflammation may prevent adverse cardiac damage. This involves the recognition of damage; activation and release of soluble mediators such as cytokines, chemokines, and proteases; and immune cells such as monocytes, macrophages, and neutrophils. This is important in the context of doxorubicin-induced cardiotoxicity as well. Doxorubicin (Dox) is an effective chemotherapy against multiple cancers but at the cost of cardiotoxicity. The innate immune system has emerged as a contributor to exacerbate the disease. In this review, we discuss the current understanding of the role of innate immunity in the pathogenesis of cardiovascular disease and dox-induced cardiotoxicity and provide potential therapeutic targets to alleviate the damage.

## 1. Introduction

Cardiovascular heart diseases (CVDs) are heart and vascular disorders and include coronary heart disease, cerebrovascular disease, peripheral arterial disease, rheumatic heart disease, and congenital heart disease. CVDs are the leading cause of morbidity and mortality globally [1]. According to WHO, 17.9 million people died of CVDs in 2019, which represents 32% of global deaths. In the United States alone, about 697,000 deaths occurred due to CVD in 2020, which accounts for one in every five deaths [2,3]. CVDs also represent a major economic burden on healthcare systems that costs over USD 300 billion each year [4]. Studies have identified key molecular mechanisms leading to the development of therapeutic drugs. A growing body of evidence suggests that immune cells are key players in the development of CVDs and, in particular, inflammation triggers the early phases of the CVDs including the atherosclerotic process and myocardial infarction [5,6]. Indeed, an increase in inflammatory cytokines is associated with an increased risk of developing CVDs. The CANTOS trial (Canakinumab Anti-inflammatory Thrombosis Outcome Study) demonstrated the key role of innate immune cells in CVDs [7]. The study was the first to demonstrate that anti-inflammatory treatment by blocking pro-inflammatory IL-1β significantly reduced systemic inflammation and lowered the rate of recurrent cardiovascular events and cardiovascular death in patients with previous myocardial infarction (MI) [7].

The anthracycline drug doxorubicin is a highly effective anti-cancer chemotherapy. Despite the fact that other anthracyclines have been developed, Doxorubicin (Dox) is widely prescribed to treat leukemia, lymphoma, Ewing sarcoma, osteosarcoma, neuroblastoma, and breast cancer [8]. Studies suggest Dox intercalates DNA and disrupts topoisomerase-II-mediated DNA repair. Furthermore, it generates free radicals and induces damage to the cell membrane, DNA, and proteins [9]. The clinical use of Dox is compromised by cardiac dysfunction that often progresses to heart failure. In contrast to the general population, young adult and childhood cancer survivors are at significant risk of developing cardiac failure [8]. Accumulating evidence suggests that Dox induces inflammatory responses via innate immune cells that progressively result in cardiac damage. In this review, we discuss the role of innate immune cells, particularly neutrophils and macrophages, along with inflammatory mediators including cytokines and chemokines in CVDs and Dox-induced cardiotoxicity.

## 2. Inflammation in the Heart

Inflammation that occurs due to trauma or chemically induced injury and in the absence of pathogens is termed ‘sterile inflammation’. In the case of cardiac injury/damage, sterile inflammation forms the foundation of the first phase of cardiac remodeling and involves the production of chemokines and cytokines and the recruitment of innate immune cells such as neutrophils and macrophages [10]. During tissue damage and inflammation, endogenous molecules related to injury are released as damage-associated molecular patterns (DAMPs). Innate immune cells recognize DAMPs via certain pattern-recognition receptors (PRRs) including Toll-like receptors (TLRs), nucleotide-binding and oligomerization domain (NOD)-like receptors, and C-type lectin receptors. Once these danger signals are detected, an inflammatory cascade is activated. This cascade includes the release of cytokines/chemokines, the activation of inflammatory pathways, and the subsequent recruitment of immune cells from circulation and bone marrow. The inflammatory phase promotes the clearance of necrotic cells and damaged tissues and is followed by a reparative phase of tissue repair that involves the deposition of a new extracellular matrix (ECM). This biphasic process is well regulated for efficient tissue repair. However, unchecked activation and chronic inflammation are detrimental and lead to aberrant cardiac damage.

Cardiac remodeling post ischemic injury occurs in three phases: inflammation, granulation, and maturation. The inflammatory phase is characterized by an increase in pro-inflammatory cytokines and chemokines, such as interleukin 1 (IL-1), IL-6, IL-8, tumor necrosis factor (TNFα), granulocyte colony-stimulating factor (G-CSF), GM-CSF, and CXCL-1. This leads to the recruitment of neutrophils and subsequently macrophages, two essential components of the myeloid system [11]. The granulation phase involves ECM turnover and the differentiation of cardiac fibroblasts and the final maturation phase includes deposition of a new ECM including collagen. Excessive deposition of collagen, however, can lead to fibrosis, and subsequently heart failure. Cardiac fibrosis is a characteristic of several conditions, such as cardiomyopathy, MI, pressure overload, and the aging process [12]. Below we have discussed the role of the innate immune system in heart post damage (Figure 1).

### 2.1. Pattern-Recognition Receptors

Toll-like Receptors

In patients with MI, endogenous DAMPs are released from damaged cells and detected by TLRs and other PRRs, which initiate a signaling cascade. These DAMPs include: heat-shock proteins, S100 proteins, uric acid, high mobility group box protein 1 (HMGB1), and endogenous nucleic acids. Additionally, components of the ECM, such as hyaluronan, heparan sulfate, and proteoglycans, can also act as DAMPs [10]. TLRs comprise three structural domains: an extracellular C-terminal leucine-rich repeat domain, a central transmembrane domain, and a cytoplasmic domain. TLR signaling occurs via two main pathways: the Myeloid differentiation primary response 88 (MyD88)-dependent and TRIF-dependent pathways.

In the MyD88-dependent pathway, there is first a recruitment of a Toll/IL-1 receptor domain containing an adaptor protein that initiates TLR4/2-related signaling. This leads to activation of the downstream IL-1 receptor-associated kinase 4 (IRAK4), induction of IRAK1 phosphorylation, and recruitment of TNF-receptor-associated factor-6 (TRAF6). Phosphorylated TRAF6 forms a complex with transforming growth factor β-activated kinase (TAK- 1), TAK1-binding protein-1 (TAB1), and TAB2. This complex then interacts with ubiquitin ligases to activate the TAK1. Next, activated TAK1 phosphorylates the inhibitor of nuclear factor-κB (IκB) Kinase (IKK) complex and p38 kinases. The IKK complex will then phosphorylate I-κB. I-κB’s subsequent ubiquitylation and degradation leads to NF-κβ translocation to the nucleus, which in turn leads to pro-inflammatory cytokine production [13].

TLRs also signal through the TIR-domain-containing adapter inducing the interferon-β (TRIF)-dependent pathway, which involves signaling via the TRIF-related adaptor molecule (TRAM) and TRIF. This signaling activates TANK Binding Kinase-1 (TBK1) and the eventual activation of interferon-regulatory factor-3 (IRF3). TRIF-dependent signaling can also occur through NF-κβ via TRAF6 recruitment, as described above [14].

TLRs in the heart

TLRs have been shown to play an important role in immune signaling in the heart after damage. In the heart, TLR2, -3, -4 and -6 have been identified in cardiomyocytes while TLR1 through 6 have been found in smooth muscle and endothelial cells [15]. In particular, TLR2 and TLR4 are important in the inflammation of the heart. TLR4-deficient mice have a weakened inflammatory response characterized by reduced levels of NF-κβ and pro-inflammatory cytokines TNFα and IL1β, smaller infarctions, and fewer cardiac infiltrations [16]. TLR2 activation in the heart has been shown to upregulate pro-inflammatory cytokines. At first this is beneficial as it leads to mitochondrial stabilization; however, sustained TLR signaling leads to the activation and recruitment of leukocytes to the cardiac microenvironment, which in turn leads to tissue destruction [12].

Inflammasome

The activation of the NF-κβ pathway can lead to the expression of inflammasome components. In particular, the NOD-, LRR-, and pyrin-domain-containing protein 3 (NLRP3) inflammasome has been shown to be induced in cardiovascular diseases. This cytosolic protein consists of three domains: a C-terminal leucine-rich repeat domain, a NOD also called a NACHT domain, and an N-terminal pyrin domain. NLRP3 binds to an adaptor protein, apoptosis-associated speck-like protein containing a caspase recruitment domain: apoptosis-associated speck-like protein containing a CARD (ASC). ASC binds to pro-caspase 1 and activates it. The active caspase 1 cleaves its substrates including pro-IL-1β and pro-IL-18. This leads to the extracellular release of mature IL-1β and IL-18. These two cytokines have been implicated in several cardiovascular diseases (CVDs) including atherosclerosis, hypertension, and myocardial infarction [17,18]. Inflammasomes also play a role during granulopoiesis, aiding in myeloid lineage specification. Specifically, caspase-1 cleaves transcription factor GATA1, thus increasing the production of neutrophils. Furthermore, IL-1β activates a PU.1-dependent gene program that stimulates granulocyte lineage differentiation [19].

### 2.2. Cytokines

Inflammatory cytokines and chemokines play an important role in the pathogenesis of myocardial dysfunction and cardiac remodeling. These pleiotropic, multi-functional cytokines are upregulated in response to myocardial injury in patients following MI [20]. The early inflammation phase of remodeling is characterized by the release of TNFα, IL-1α, IL-1β, and IL-18. The next phase is characterized by the release of anti-inflammatory cytokines such as TGFβ.

Pro-inflammatory cytokines

High TNFα levels have consistently been documented in experimental models of heart failure and in patients. This cytokine exerts a negative inotropic action on cardiomyocytes by disturbing the calcium homeostasis, triggering apoptosis by activating cell death pathways [20]. In fibroblasts, the balance between matrix metalloproteinases (MMPs) and their inhibitors is affected by TNFα, leading to extracellular (ECM) destruction [21]. In the microvasculature, TNFα modulates cyclooxygenase-2, which in turn leads to increased expression of intercellular adhesion molecule 1 (ICAM-1) and vascular cell adhesion molecule 1 (VCAM-1). This causes pro-inflammatory immune cells to migrate to and remain in the cardiac microcirculation, leading to tissue injury and cardiac dysfunction [22].

IL-1 has two ligands, IL-1α and IL-1β, both of which have high sequence homology. Infarcted hearts have been shown to have increased levels of IL-1 family proteins. It has been observed that IL-1 has a pro-apoptotic and hypertrophic effect on cardiomyocytes, which can depress cardiac contractility. IL-1, along with TNFα, causes cardiomyocyte apoptosis via pathways involving nitric oxide (NO) and by upregulating Bax, Bak, and caspase-3. In addition, IL-1 has been shown to promote the expression of MMP-3, MMP-8, and MMP-9 on cardiac fibroblasts while downregulating the expression of their inhibitors’ tissue inhibitor of metalloproteinase (TIMP-2 and TIMP-4), which could lead to the degradation of the ECM and is detrimental to heart tissue [23].

Anti-inflammatory cytokines

Transforming growth factor β (TGFβ) is crucial in cardiac fibrosis and remodeling. TGFβ works by binding to Ser/Thr kinase receptors—TGFβ receptors type I (TGFβRI) and type II (TGFβRII)—on the surface of cells. Macrophages that infiltrate following myocardial injury have been observed to release TGFβ in significant quantities. Once released, TGFβ induces the expression of genes that promote and increase ECM production, which in turn suppresses MMP expression and enhances cardiac repair. Additionally, TGFβ binds to its type I and II receptors and initiates SMAD signaling. This signaling sets into motion the transformation of fibroblasts to myofibroblasts and eventually ECM deposition in a cyclical manner. TGFβ can also contribute to fibroblast differentiation by combining with other pro-inflammatory cytokines to promote endothelial-to-mesenchymal differentiation [24].

Chemokines

Chemokines are a family of small chemotactic cytokines that regulate the transmigration of immune cells to target tissues. Chemokines interact through G-protein-coupled chemokine receptors. Chemokines play a critical role in migrating leukocytes to sites of inflammation. Furthermore, chemokines may play a role in CVD by affecting the activation of leukocytes, survival of monocytes, foam cell formation, and thrombus formation as well as lymph-angiogenesis.

The chemokine C-X-C motif chemokine ligand 1 (CXCL1) has been shown to play a critical role in CVD and heart failure [25]. CXCL1 acts as the chief chemoattractant for neutrophils but can also recruit monocytes. CXCL1 acts through its receptor CXCR2, which is present on neutrophils [25]. In addition to its chemotactic role, CXCL1 can induce cardiac fibrosis via TGF-SMAD 2/3 signaling [26].

In addition to CXCL1, C-C motif chemokine ligand 2 (CCL2), also known as the monocyte chemoattractant protein-1 (MCP-1) axis, plays an important role in the recruitment of classical monocytes (pro-inflammatory CD14^+^CD16^−^ monocytes) and the development of CVD. CCL2 is secreted by immune cells, smooth muscle cells, endothelial cells, and fibroblasts. The CCL2-CCR2 axis regulates the transmigration of a wide range of immune cells including monocytes, macrophages, T cells, and NK cells. Plasma levels of CCL2 have been shown to be associated with an increased risk for heart failure, atherosclerosis, and coronary heart disease [27]. In a preclinical model, CCL2-CCR2 was identified as being crucial in the development of atherosclerosis such that the selective deletion of CCR2 significantly decreased atherosclerotic lesions and was associated with a reduced accumulation of monocytes/macrophages. Furthermore, CCR2 deficiency reduced Ly6C^high^ monocyte infiltration to the site of infarction and inhibited inflammation. Reduction in inflammatory monocytes promoted myocardial infarct healing.

### 2.3. Neutrophils in Heart Damage

Neutrophils are the most abundant granulocytic leukocytes in human peripheral blood (50–70%). These cells are an essential part of the innate immune system and help with host defense and inflammation resolution. They are among the shortest-lived cells and require continuous replacement from the bone marrow. Neutrophils are generated by a process known as granulopoiesis. This starts with a granulomonocytic progenitor that goes through maturation stages from myeloblast to promyelocyte to myelocyte and then a mature neutrophil. These stages are regulated by the cytokine G-CSF, which also promotes the circulation of neutrophils by disturbing the CXCR4-CXCL12 interaction that helps keep neutrophils in the bone marrow. G-CSF causes the release of neutrophils by reducing the expression of CXCL12 on stromal cells in the bone marrow and CXCR4 on the neutrophils [27]. Once tissue damage occurs, and inflammatory stimuli and chemokines are released, neutrophils home into these signals by leaving the circulation and migrating to the site of damage.

Neutrophils in CVDs

Neutrophils play a detrimental role in acute MI and neutrophil count correlates with infarct size and heart failure development [28]. Neutrophils have also been shown to be involved in atherosclerosis, thrombosis, and acute coronary syndrome.

Atherosclerosis is one of the most common causes of MI leading to death. Neutrophils have been shown to promote the initiation of atherosclerosis by dysregulating vascular endothelial cells. Moreover, increased neutrophil counts have been observed in human atheroma specimens [28]. In animal models with atherosclerotic plaques, activated macrophages produce chemokines to attract neutrophils. The neutrophil transmigration is mediated by oxidized low-density lipoproteins via an increase in endothelial contractility and upregulation of ICAM-1 [29]. Depletion of neutrophils reduced atherogenesis. In atherosclerotic lesions, neutrophils release CAMP and AZU1, which recruit inflammatory monocytes and help upregulate the expression of adhesion molecules on the surface of endothelial cells, regulating the endothelial cell layer permeability [19,30]. Myeloperoxidase released by neutrophils catalyze the conversion of hydrogen peroxide to hypochlorous acid, which produces toxic chloramines that can alter lipoprotein function. Neutrophils also release reactive oxygen species (ROS), which contribute to plaque vulnerability by oxidizing LDL and recruiting additional neutrophils in atherosclerotic lesions [31]. Further, a study suggests that neutrophils co-localize with TLR-2-expressing atherosclerotic plaques on endothelial cells. TLR-2 activation on endothelial cells leads to cell stress and apoptosis. This effect is substantially enhanced by the adhesion of neutrophils to endothelial cells in atherosclerotic lesions. This in turn correlates with an increased number of luminal apoptotic endothelial cells in atherosclerotic lesions [19].

In MI, neutrophils can exacerbate damage and increase infarct size. Neutrophils infiltrate at the ischemic border zone and release ROS, which then leads to acute inflammation and cardiomyocyte apoptosis, leading to the destruction of cardiac tissue and ultimately impaired heart function in hearts. Additionally, the release of myeloperoxidase into extracellular space causes the generation of cytotoxic aldehydes, oxidative stress, and the activation of enzymes that degrade the ECM, leading to impairment in remodeling. During acute MI, neutrophils release Ca^2+^-binding proteins S100A8/A9 that prime the NLRP3 inflammasome to release IL-1β, which stimulates granulopoiesis in the bone marrow. This leads to the accumulation of neutrophils in the infarcted heart, adverse remodeling, and heart failure [32]. Anti-inflammatory strategies have focused on inhibiting cardiac neutrophil recruitment. Brahma-related gene 1 (BRG1) is crucial in mediating neutrophils’ adhesion to the endothelium infiltration of the infarct [33]. When BRG1 was deleted, reduced infarct size, less fibrosis, and recovery of cardiac function were reported. Additionally, obstructing the interaction between CC motif chemokine 5 (CCL5) and CXC motif chemokine receptor 4 (CXCR4) in MI shrank infarct size, conserved heart function, and was related to a reduced presence of neutrophils and the formation of neutrophil extracellular traps [34]. A study conducted in a mouse model of MI found that starting 3 days after MI, two populations of neutrophils were found in the heart: SiglecF^hi^ and SiglecF^low^. SiglecF^low^ represented young blood neutrophils whereas SiglecF^hi^ shared characteristics of aged neutrophils (i.e., low CD62L, high CXCR4, expression of specific transcripts, enrichment of ribosomal protein encoding genes). The SiglecF^hi^ population was also seen to express high surface ICAM1 expression. As a result, these subsets have different functional capacities. The SiglecF^hi^ population was shown to have higher phagocytic capacity and higher ROS production than the SiglecF^low^ population and this was detrimental to cardiac remodeling [35].

Neutrophil Elastase

Neutrophil elastase (NE) is a 29 kDa serine protease from the chymotrypsin family found in the primary granules of neutrophils. NE activity depends upon a catalytic triad comprising aspartate, histidine, and serine residues that are separated in primary granules but come together at the enzyme’s active site in the tertiary structure. Primarily, NE is active in neutrophils’ response against bacteria, but it has also been known to cause ECM destruction. In the granules, an acidic milieu protects the cells from proteolytic activity. Upon activation of neutrophils, fusion between granules and cytoplasmic phagosomes occurs causing alkalosis, which further triggers degranulation. NE is liberated into extracellular space in both free and membrane-bound forms [36].

NE can target ECM, cell-surface ligands, proteins, and adhesion molecules. NE can degrade matrix proteins, including collagen, fibronectin, proteoglycans, heparin, and fibrin. NE also has the ability to evade tissue anti-proteases in the extracellular environment [37].

Neutrophil elastase in CVDs

As discussed previously, NE is in the primary granules of neutrophils and is released during degranulation. Continuous NE secretion can cause excessive tissue destruction. NE has also been implicated in arthritis, respiratory, and cardiovascular diseases. Patients with acute MI have elevated levels of NE in their plasma. NE degrades elastin, collagen, and fibrinogen, which can lead to damage following MI. Additionally, NE induces IL-6 release which leads to the impairment of cardiac contractility [38]. Investigations in NE ^-/-^ mice confirmed that NE was responsible for promoting excessive inflammation and activating pro-inflammatory cytokines that contributed to the cardiac damage following MI [39]. NE enhanced myocardial injury by suppressing the phosphatidylinositol-3 kinase (PI3K)/protein kinase B (AKT) pathway. Insulin-driven AKT signaling is critical in preventing apoptosis [40]. NE can enter intracellular space and mediate the degradation of the insulin receptor substrate protein, which in turn prevents AKT activation and protection against apoptosis. In NE^-/-^ mice, AKT signaling was activated and cardiomyocytes were protected from apoptosis. In this study, a pharmacological inhibitor of NE, silvestat, was found to improve survival and cardiac function after MI [39].

In a study conducted in patients with type II diabetes and MI, there were elevated levels of NE and neutrophil in atherosclerotic plaques. Additionally, an association between high NE and MI severity was observed [41]. The study showed that higher NE was detected in detached emboli of atherosclerotic plaques, contributing to plaque instability [41]. NE has also been found to mediate a chronic inflammatory state by the activation of a pro-form of TNF and IL-1β, PAR2, and phospholipase C, leading to the translocation of NF-κβ and activation of that signaling pathway [42,43]. NE can also lead to pro-apoptotic signaling by endothelial cells via extracellular signal-regulated kinases (ERK), c-Jun N-terminal kinases (JNK), and p38 mitogen-activated protein kinases (MAPK), which can lead to the apoptosis of cells [44].

### 2.4. Macrophages in CVD

In addition to neutrophils, macrophages are immune cells that contribute to innate immunity. Resident macrophages derived from embryonic or fetal liver tissue, and infiltrating macrophages are key regulators of normal tissue homeostasis, regeneration, and repair. In addition to immune activation, macrophages play a central role in tissue damage and repair through functions including phagocytosis of cell debris, inflammatory cell recruitment, and regulation of neovascularization and fibrosis [45,46]. Furthermore, macrophages perform tissue-specific functions. For instance, microglia are critical for neuronal development and cardiac macrophages regulate vascular and cardiac regeneration as well as facilitate electrical conduction in the heart [45,47,48].

Cardiac macrophage heterogeneity and diversity

Genetic lineage tracing studies suggest that multiple macrophage populations with distinct ontological origins reside within the heart. Tissue-resident macrophages derived from the embryonic yolk sac or fetal liver populate the heart at distinct stages of development and self-maintain their population by local proliferation [46,49,50]. Macrophages derived from circulating monocytes also reside in the heart. Studies suggest that monocyte-derived macrophages can replenish tissue-resident macrophages that have been depleted due to cardiac damage or aging [50]. Both resident and monocyte-derived macrophages play a distinct role and coordinate tissue homeostasis and repair.

Macrophages in the adult mouse heart can be classified based on CCR2 and MHC-II expression as non-classical monocyte-derived macrophages (CCR2^−^MHC-II^low^ and CCR2^−^MHC-II^high^) or classical monocyte-derived macrophages (CCR2^+^MHC-II^high^) [50,51]. CCR2^−^MHC-II^low^ and CCR2^−^MHC-II^high^ are tissue-resident macrophages [50,52]. They maintain their populations independently of circulating blood monocytes through local proliferation. In contrast, CCR2^+^MHC-II^high^ macrophages are derived from circulating blood monocytes. In general, CCR2^−^ resident macrophages are considered anti-inflammatory with minimal ability to induce inflammatory cytokines and chemokines in response to injury [45,46]. Studies suggest that these resident macrophages are reparative in nature and regulate angiogenesis and stimulate cardiomyocyte proliferation [50]. In contrast, CCR2^+^ macrophages are pro-inflammatory, and induce a strong inflammatory cytokine response to LPS. Infiltrating CCR2^+^ macrophages induce neutrophil recruitment to the site of injury and site of inflammatory cytokine production and contribute to collateral myocardial damage [50,52].

Role of macrophages in cardiac damage and repair

Macrophages play a critical role in both cardiac damage and repair. While macrophages regulate tissue regeneration and repair by secreting growth factors, they can also worsen tissue injury and impair repair by producing ROS and inflammatory cytokines [45]. Cardiac injury due to various pathophysiological processes results in the release of pathogen- or damage-associated molecular patterns (PAMPs or DAMPs). DAMPs, including adenosine triphosphate (ATP), high-mobility group protein B1 (HMGB1), heat-shock proteins (HSPs), and S100A8/9 are released by necrotic cell death. Resident macrophages express recognition receptors such as TLRs, mannose receptors, and purinergic receptors to recognize DAMPs, which leads to an initiation of an inflammatory cascade [47].

Cardiac macrophages follow a biphasic response to cardiac injury and repair. The initial pro-inflammatory response is characterized by the activation of macrophages and the production of inflammatory chemokines and cytokines (IL-1β, IL-6 and TNF-ɑ) followed by the infiltration of monocytes and neutrophils [46,47,50]. Monocytes (Ly6C^high^) differentiate into pro-inflammatory macrophages (CCR2^+^) in response to the local inflammatory environment and outnumber resident macrophages. Furthermore, macrophages release MMPs that disrupt tissue matrices and facilitate the infiltration of leukocytes into the area of injury [45,46,53]. Monocytes and macrophages function as scavenger cells and undergo extensive phagocytosis to clear cellular debris. This is characterized by increase in monocyte-derived CCR2^+^Ly6C^high^ (M1-like) macrophages and decrease in CCR2^−^Ly6C^lo^ (M2-like) macrophages. The early inflammatory phase is quickly followed by an increase in M2-like reparative macrophages that promote wound healing. These macrophages secrete numerous growth factors including platelet- derived growth factors (PDGFs), insulin-like growth factor-1 (IGF-1), vascular endothelial growth factor (VEGF) and TGFβ [54], which promote cellular proliferation and neoangiogenesis [46]. These reparative M2-like macrophages play a critical role in dampening inflammation by secreting anti-inflammatory mediators (IL-10 and TGF-β) and upregulate the expression of cell-surface ligands (PDL1 and PDL2) [55]. In addition, studies suggest secreted TGF-β mediates the differentiation of fibroblasts into myofibroblasts. Myofibroblasts are recruited for wound healing during cardiac injury by tenascin-C secreted by macrophages [55].

The exact role of M1/M2 macrophages in cardiac injury and repair continues to be debated. M1 macrophages have been shown to promote cardiac injury [56]. while M2 macrophages are important for repair [57]. This is furthermore complicated by the classical macrophage classification that relies on cell-surface markers and inflammatory state [51,52]. This classification system is now recognized to be imperfect due to macrophage plasticity and their ability to change from M1 to M2 and M2 to M1 thereby changing the expression of M1/M2 markers in response to the microenvironment. Macrophage function must be tightly regulated and orchestrated to mediate effective cardiac repair, which requires the efficient removal of cellular debris, a strong and transient inflammatory response, the subsequent secretion of growth factors, neovascularization, myofibroblast infiltration, and scar formation. Indeed, disturbances at any stage of the process can lead to abnormal repair. Inefficient clearance of cellular debris, uncontrolled production of inflammatory mediators and growth factors, or deficiencies in anti-inflammatory reparative macrophages contribute to a chronic wound that ultimately leads to the formation of pathological fibrosis. Studies targeting phagocytosis receptors in macrophages such as MerTK and Mfge8 suggest inefficient cellular debris clearance results in impaired wound healing and a decline in cardiac function [58,59]. Furthermore, in a study, it was shown that targeting macrophage polarization yielded efficient cardiac healing in mouse models of MI and cardiac pressure [60]. After cardiac injury, mice with the specific deletion of the transcription factor GATA3 (mGATA3KO) in myeloid cells had improved cardiac function, less left ventricular (LV) dilation, reduced scarring, and improved contractility following MI. This effect was associated with an increase in CCR2^+^Ly6C^high^ (M1) and decrease in CCR2^−^Ly6C^low^ (M2) macrophages suggesting that M2 macrophages may contribute to reduced cardiac fibrosis and remodeling [60]. Another study by Leor et al. showed that early intracardiac administration of activated macrophages after MI improved vascularization and myofibroblast accumulation and reduced ventricular dysfunction [61]. Taken together, these findings indicate that there is an optimal balance between M1 and M2 macrophages in the response to cardiac insult and damage and that immune modulatory approaches targeting cardiac macrophages may be an efficient strategy for promoting myocardial repair and function following injury.

## 3. Dox-Induced Cardiomyocyte Injury

Dox-induced cardiomyocyte cell death is complex, and is regulated by multiple mechanisms of action (Figure 2) [62,63,64]. The major mechanisms leading to cardiomyocyte cell death include (1) the generation of reactive oxygen species (ROS) and nitrogen species (RNS) that leads to protein and DNA damage and lipid peroxidation; (2) the inhibition of topoisomerase-II (TOP-IIβ) resulting in DNA strand breaks; and (3) impaired mitochondrial function and the disruption of the electron transport chain (ETC). This leads to regulated or unregulated cell death apoptosis, ferroptosis and pyroptosis as well as necroptosis and the eventual release of inflammatory mediators [65].

Several mechanisms lead to the generation and accumulation of ROS/RNS leading to oxidative stress in cardiomyocytes. Dox can directly bind to endothelial nitric oxide synthase (eNOS) and generate Dox-semiquinone radical. This generated superoxide (O_2_^−^) radical. Furthermore, Dox increases the levels of inducible nitric oxide synthase (iNOS) and nitrotyrosine (NT). Superoxide and NT generate other potent oxidants that eventually cause apoptosis [65]. In addition, cationic Dox can form a complex with free iron and form a Dox-Fe complex. The complex alters iron metabolism and directly interacts with free oxygen to generate ROS as well as enhances lipid peroxidation [64,65]. This effect is enhanced by the upregulation of transferrin (TfR), allowing the transport of more iron into the cell leading to excess intracellular iron. This is furthermore complicated by the mitochondrial accumulation of iron in mitochondria. Dox downregulates ABCB8 protein, responsible for exporting iron outside the mitochondria, as well as inhibits mitochondrial ferritin thus disrupting the homeostasis of mitochondrial iron [65,66]. The accumulation of Dox inside mitochondria is further enhanced by high-affinity binding to cardiolipin. Cardiolipin-bound Dox disrupts complexes I, III, and IV, which are essential in the electron transport chain (ETC) for generating ATP. Overall, the accumulation of Dox in mitochondria leads to enhanced ROS/RNS production, lipid peroxidation, DNA and protein damage, loss of ATP, and mitochondrial permeability [63].

Dox also causes calcium dysregulation inside the cell, calcium overload, and induces apoptosis [63,64,65]. Doxorubicinol, also known as DOXOL, is the hydroxyl metabolite of Dox that affects calcium homeostasis. DOXOL modulates calcium ATPase in sarco/endoplasmic reticulum (SER). It also disrupts the sodium/calcium exchange channel in SER, regulating the calcium level, which plays a critical role in cardiomyocyte contractility. Calcium overload activates calcium-dependent proteases such as calpains. Activated calpains cleave caspase-12 and induce the apoptosis of cardiac cells [64]. Further, calcium overload activates calcium-dependent CaMKII (calmodulin activated protein kinase II) and PLN (phospholamban), which generate peroxides, activates caspase 3/9, and enhance apoptosis [63,64].

Dox also induces cardiomyocyte apoptosis by both intrinsic and extrinsic mechanisms. Dox treatment causes oxidative stress by excess ROS/RNS, as described above, as well as mitochondrial damage. These events cumulatively cause the swelling of mitochondria, loss of membrane potential, and release of cytochrome c and apoptosis-inducing factor (AIF) in the cytosol. This leads to the activation of caspase 3/9 resulting in apoptosis and cell death [63,65]. Further, oxidative stress activates HSF-1 (heat-shock factor-1) and stabilizes p53 protein. This leads to the upregulation of pro-apoptotic factors such as Bax, and downregulation of anti-apoptotic factors such as Bcl-XL [64]. In addition, Dox also induces the upregulation of death receptors (DRs) on the cell surface. Death receptors such as TRAIL-receptor (tumor-necrosis-factor-related apoptosis-inducing ligand), Fas, DR4, DR5, and TNFR (tumor necrosis factor receptor), when bound to their cognate ligand, trigger the activation of caspase cascade ultimately leading to apoptosis [63,65].

Another important cellular target of Dox is topoisomerase-II. Adult cardiomyocytes abundantly express Topoisomerase-IIß (TOP-IIß). TOP-IIß complexes with Dox and DNA to generate a ternary cleavage complex. The ternary complex induces single- and double-strand breaks in mitochondrial and nuclear DNA. DNA damage caused by Dox can lead to the overexpression of p53, leading to an increase in the expression of proapoptotic targets thus activating cell death. Furthermore, the ternary complex can lead to the inhibition of mitochondrial biogenesis and gene expression leading to the inhibition of secondary oxidative phosphorylation [66,67].

### 3.1. Role of Immune Response in Dox-Induced Cardiotoxicity

Although studies have demonstrated the critical role of the innate immune system in the heart, the role of the innate immunity system in Dox-induced cardiotoxicity has not been assessed in great detail. The limited studies conducted in this field have identified some key components of the innate immune system that contributed to acute Dox-induced heart damage. Here, we discuss the role of immune cells and cytokines in the context of Dox-induced cardiotoxicity.

#### 3.1.1. Cytokines/Chemokines

Several cytokines and chemokines have been targeted with respect to Dox-induced cardiotoxicity. In one study, IFN-γ was implicated in Dox-induced cardiotoxicity [68]. IFN-γ was shown to reprogram lipid metabolism and sensitize cardiomyocytes to cardiotoxicity, which worsened heart function. Cardiomyocytes need fatty acids to develop respiratory capacity and impeding oxidation will interfere with that process. AMP-activated protein kinase (AMPK) signaling enhances fatty acid oxidation and helps regulate the respiratory capacity of cardiomyocytes. It was observed that, with Dox treatment, IFN-γ interfered with AMPK signaling by the suppression of the AMPK/ACC axis in a p38-dependent pathway, which enhanced the Dox-induced cardiotoxicity [68]. Importantly, antibody treatment against IFN-γ improved the heart function in mice. This demonstrated that inhibiting IFN-γ could mitigate new as well as previously established Dox-induced cardiotoxicity. The investigators also found that IFN-γ inhibition had no effect on the therapeutic efficacy of Dox in mice with tumors [69].

A study in breast cancer patients receiving Dox found that the plasma levels of cytokines CCL27 and macrophage migration inhibitory factor (MIF) were elevated after two cycles of Dox [70]. CCL27 is associated with the homing of T lymphocytes to sites of inflammation whereas MIF is a crucial cytokine involved in acute and chronic inflammatory response. MIF has been found to play a role in maintaining cardiac homeostasis and found to be elevated in MI, atherosclerosis, and other disorders [71]. MIF could play a role in protecting against cardiotoxicity by attenuating the loss of autophagy and ATP availability in the heart leading to the maintenance of cardiac homeostasis [72]. CCL23, also called macrophage inflammatory protein 3, was also found to be elevated after each cycle. This cytokine has a suppressive effect on hematopoietic progenitor cells. Previous studies have shown an association between high levels of CCL23 and coronary atherosclerosis [70]. Another study, conducted in HER2^+^ breast cancer patients receiving anthracycline, revealed a significant increase in CXCL10 levels from baseline to post-anthracycline and post-trastuzumab treatment. This increase correlated with a decline in global longitudinal strain [68]. CXCL10 has several roles, including serving as a chemoattractant for monocytes, macrophages, T cells, and NK cells and promoting T-cell adhesion to endothelial cells, thereby leading to the significant infiltration of these immune cells during cardiac remodeling [73].

When aortas of mice treated with Doxorubicin were studied, there was a higher concentration of pro-inflammatory mediators such as IL-1β, IL-2, IL-6, and TNFα. TNFα levels had the highest elevation, which was associated with intrinsic wall stiffness that was prevented by the inhibition of TNFα [74].

A study conducted in breast cancer patients who received Dox found that compared to baseline the levels of IL-10 were significantly increased 7 days after therapy completion in patients with cardiotoxicity. Increased levels of plasma NT-proB-type Natriuretic peptide (NT-proBNP, a marker for cardiac injury) correlated with the increased IL-10 levels in patients with cardiotoxicity. IL-10 levels were also positively correlated with IL-1β in the patients with cardiotoxicity, even though IL-10 is an immunosuppressive cytokine [75].

#### 3.1.2. TLRs

TLRs have also been found to be important as part of the innate immune response to Dox-induced cardiotoxicity [76]. TLR5 was found to be significantly elevated in the hearts of mice treated with Dox. TLR5 deficiency led to reduced NADPH oxidase 2 (NOX2) levels in particular. NOX2 is an isoform of NADPH oxidase, a primary source of ROS in the heart. This was important as the investigators demonstrated that TLR5 activated NOX2 through Syk phosphorylation [76]. TLR5 deficiency attenuated this effect. Dox was found to activate the p38 signaling pathway, which led to the apoptosis of cardiomyocytes. This p38 pathway was NOX2-dependent and hence activated by ROS. This pathway was also inhibited in TLR5-deficient mice [76,77]. TLR5 deficiency led to lower TNFα and IL-1β mRNA levels and NF-κβ translocation was also inhibited in these mice and this led to improvements in heart function and less myofibrillar disruption in mice treated with Dox [76].

In a study conducted in mice with TLR9 deficiency that was treated with Dox, it was found that cardiac function, myocardial fibrosis, and markers for myocardial damage were all reduced as compared to wild-type (WT) mice treated with Dox alone. TUNEL staining further revealed that in TLR9 KO mice with Dox treatment there was a significantly reduced number of apoptotic cardiomyocytes and reduced ROS production compared with wild-type (WT) mice with Dox treatment. Furthermore, it was found that TLR9 promoted the oxidative stress and apoptosis through p38 MAPK-dependent autophagy leading to the death of cardiac cells [78].

Another TLR whose relationship to Dox-induced cardiotoxicity has been studied in mice is TLR2. TLR2-KO mice showed less NF-κβ activation, along with a lower production of pro-inflammatory cytokines (TNFα and IL-6), compared with WT mice. The TLR2-KO mice had higher survival rates than WT mice after Dox treatment. Furthermore, fewer TUNEL-positive cells were found in the myocardium, and caspase-3 activation was suppressed in the TLR2 KO mice with Dox treatment [79]. In a study measuring inflammatory biomarkers in patients with heart failure, expression levels of TLR2 increased in patients in both the Dox group without heart failure and the Dox plus heart failure group [80]. In another study evaluating the anti-inflammatory role of LCZ696 (sacubitril/valsartan), an angiotensin receptor neprilysin inhibitor that is used to reduce the risk of cardiovascular death for patients with heart failure, with respect to TLR2 deficiency, it was found that administration of the drug improved heart function and prevented cardiac fibrosis after Dox treatment. In addition, LCZ696 also prevented high TNFα expression. In TLR2-KO mice, similar results were observed, suggesting a connection between drug action and TLR2. Further studies found that LCZ696 attenuated the formation of the TLR2-MyD88 complex and this in turn alleviated the negative effects of Dox, as Dox promotes the formation of the TLR2-MyD88 complex [81].

TLR4, a receptor of endotoxin, has also been shown to contribute to cardiac inflammation in Dox-induced cardiotoxicity. TLR4-KO mice had improved LV function and a reduction in cardiac ET-1, which contributes to heart failure. Additionally, when lipid peroxidation and nitrotyrosine were examined as markers of oxidative stress in TLR4-KO mice treated with Dox, there was significantly reduced oxidative stress. A study of an animal model of ischemia/reperfusion also suggested that TLR4 contributed to the development of oxidative stress. Furthermore, it was observed that the infiltration of lymphocytes, monocytes, and macrophages was reduced in the TLR4-KO mice treated with Dox compared to Dox-treated WT mice [82]. Upregulation of the pro-apoptotic protein Bax was observed in WT Dox-treated mice, which was not seen in the TLR4-KO mice. These findings were confirmed by TUNEL assay where reduced apoptotic cells were seen in the TLR4-KO group. The study also found a significant upregulation of Bcl-2, an anti-apoptotic protein, in TLR4-KO mice with Dox treatment as compared to TLR4-KO mice. In a mouse study, downregulation of the GATA-4 pathway was seen in Dox-induced cardiomyopathy, and downregulation of this pathway is known to promote Dox-induced cardiotoxicity. However, in TLR4-KO mice, this downregulation did not occur, nor did the disease [82]. Another finding that supports the importance of TLR4 in Dox-induced inflammation was that TLR4 expression was increased in macrophages following Dox treatment. When TLR4 was suppressed or depleted by injecting TAK-242 or using TLR4^lps-del^ mice, lower myofibrillar disruption as compared to Dox groups was observed [83].

#### 3.1.3. Innate Immune Cells

The role of innate immune cells, especially neutrophils, has been discussed in great detail in CVDs. However, the role of these immune cells in Dox-induced cardiotoxicity needs to be looked at in greater detail.

Neutrophils

Neutrophils may also contribute to Dox-induced cardiotoxicity. In one study examining therapy-related clonal hematopoiesis following anti-tumor agents including Dox, cardiotoxicity was augmented by the infiltration and activation of neutrophils. In this study, an elevation of neutrophils was observed in cardiac tissue, which peaked at 7 days after treatment with a single bolus of Dox. When mice were transplanted with Trp53 heterozygous mutant bone marrow cells to establish a model of clonal hematopoiesis, neutrophil recruitment was higher in heart tissue compared to mice transplanted with WT cells post Dox treatment. Furthermore, when these heterozygous-Trp53-deficient neutrophils were analyzed for gene expression, these neutrophils were enriched for genes related to the inflammasome pathway (i.e., *Nlrp1b, Gbp5, Il18*) and chemokines (e.g., *Ccl25, Ccrl2, and Cxcl1*). When neutrophils were depleted in the mice with Trp53-deficient cells, an amelioration of echocardiographic parameters including fractional shortening was observed after Dox treatment indicating that neutrophil involvement is crucial for the detrimental effects of Dox [84]. In another study conducted in breast cancer patients receiving anthracyclines, a high level of plasma neutrophil extracellular traps was seen to be associated with Dox-induced cardiotoxicity [85].

Recently, we demonstrated that Dox treatment induced a significant infiltration of neutrophils into hearts 24 h after Dox therapy, which was accompanied by an acute and late decrease in cardiac function, disruption in vascular structures such as pericytes and endothelial cells, and an increase in collagen deposition, leading to fibrosis. The depletion of neutrophils prevented Dox-induced cardiotoxicity with the preservation of vascular structures and prevention of excess collagen deposition 10 weeks after therapy [86]. This effect was dependent on neutrophil elastase (NE) such that NE-KO mice treated with Dox had fewer apoptotic cardiomyocytes, preserved cardiac function, and preserved vascular structures compared to WT mice treated with Dox. Importantly, treating mice with a pharmacological inhibitor of NE (AZD9668) in conjunction with Dox significantly prevented the cardiotoxic effects of Dox. This study provided a potential therapeutic approach to mitigate the cardiac damage induced by Dox therapy. Additional studies are needed to elucidate the role of other neutrophil extracellular traps molecules in Dox-induced cardiac damage to more completely understand the role of innate immunity in the development of late cardiac morbidities in childhood cancer survivors [86].

Macrophages

Dox induces sterile inflammation and non-ischemic cardiac damage characterized by systemic increases in TNFɑ, IL-1β, and LPS [87,88]. The recognition of apoptotic cells, inflammatory mediators, and DAMPs by innate immune cells, macrophages, is critical for immune activation and resolution [89]. Dox-treated apoptotic cells release TNF, which amplifies the inflammation in a TNF-R1-dependent manner. This was shown by significantly reduced levels of LDH, TNF, and neutrophils in TNF-R1-KO mice following doxorubicin administration [87]. Interestingly, in a study conducted, it was shown that acute inflammation in response to doxorubicin is associated with the apoptosis of monocytes and macrophages. The investigators also found that the innate immune sensing of apoptotic cells is mediated by the TLR2/TLR9-MyD88 pathway and is key to the initiation of an acute inflammatory response to Dox [88]. In addition to TLR2, TLR4 plays a role in cardiac inflammatory signaling, such that TLR4-KO mice treated with doxorubicin had improved LV function, reduced cardiac apoptosis, and reduced inflammatory mediators [79,82,90]. As noted earlier, TLRs have been shown to play a major role in Dox-induced cardiotoxicity and activate innate immune cells, particularly macrophages, in response to injury and inflammation. Another study suggested a distinct role of TLR2 and TLR4 in mediating Dox-induced cardiotoxicity. An antibody-mediated blockade of TLR2 was associated with a reduced inflammatory response and attenuated Dox-induced cardiac dysfunction [91]. This was in contrast to TLR4 blockage, where neutralizing antibodies to TLR4 exacerbated cardiac tissue damage. However, TLRs are also expressed by non-immune cells, including cardiomyocytes, and ligand stimulation induces cardiomyocyte apoptosis and inflammatory response [92,93].

Our understanding of the precise role of cardiac macrophages on the pathophysiology of Dox-induced cardiotoxicity is limited. Studies in mouse models of Dox-induced acute cardiotoxicity suggest that Dox treatment enhanced the M1 macrophage population and suppressed the M2 macrophage population [45,50,51]. The M1 macrophage population has been shown to regulate the development of cardiac injury [50,52,94]. In another study, the dynamics of circulating monocyte-derived recruited pro-inflammatory M1 macrophages and reparative anti-inflammatory M2 macrophages in doxorubicin-induced cardiotoxicity were shown to play a critical role in cardiac injury and repair [95]. Using techniques, such as parabiosis, CX3C motif chemokine receptor 1 (CX3CR1)-based macrophage lineage tracing, and bone marrow transplantation, they showed that M1 macrophages may be the dominant population at the initial phase of cardiac injury, followed by a progressive increase in reparative M2 macrophages. This study suggests that resident macrophages are vulnerable to Dox insults but that the surviving resident macrophages are induced to proliferate [95]. Importantly, the proliferation of the reparative M2 macrophages was dependent on scavenger receptor A1 (SR-A1), a critical regulator of DAMP-induced macrophage proliferation [95]. In addition to pro-inflammatory mediators, M1 macrophages release pro-oxidative stress factors that sensitize cardiomyocytes to oxidative stress [96,97]. Immunomodulatory approaches to regulate the balance of M1/M2 macrophage can help in the resolution of cardiac injury. Another study by Ye and colleagues demonstrated that IL-22 is a critical regulator of macrophage differentiation in response to cardiac injury [97]. IL-22 deficiency reversed Dox-induced cardiac M1 /M2 macrophage imbalance and increased M2 macrophages. This effect was associated with reduced cardiomyocyte apoptosis, reduced cardiac vacuolization, and the improvement of cardiac function and LV tissue [97]. Furthermore, in another study conducted by the same group, they demonstrated that Dox prevented M2 macrophage differentiation and that IL-12p35 deficiency exacerbated Dox-induced myocardial injury supporting the importance of M1 macrophages in Dox-induced cardiotoxicity [98]. IL-12p35-KO promoted M1 macrophage differentiation, increased pro-inflammatory cytokines (IL-6, MCP-1, IL-1β, and IFN-γ), and reduced M2-macrophage-related anti-inflammatory cytokines (IL-4, IL-13 and IL-10) [98]. Supporting the protective role of M2 macrophages, another study showed that glabridin, an isoflavone from licorice root, prevented Dox-induced cardiotoxicity by modulating Dox-induced gut dysbiosis and colonic macrophage polarization. Glabridin treatment reduced the level of LPS, increased butyrate, upregulated M2-related genes (arginase -1, CD206), and downregulated M1 genes (iNOS and CXCL9) [99]. In a study conducted to examine the role of NLRP3, it was shown that NLRP3, despite involvement in sterile inflammation, regulates cardiac IL-10 production. NLRP3-deficient mice were susceptible to Dox-induced cardiomyopathy compared to WT mice. Mechanistically, NLRP3 regulates IL-10 production in cardiac reparative M2 macrophages, independent of IL-1ꞵ, and contributes to the pathophysiology of Dox-induced cardiotoxicity [100]. As an immunomodulatory approach, Singla et al. showed that treatment with embryonic stem cells (ESCs) or exosomes derived from embryonic stem cells can promote M2 reparative macrophages and reduce inflammation-induced pyroptosis. This effect correlated with an increase in M2 macrophages and improved cardiac function [101]. Furthermore, Liu et al. showed that the adoptive transfer of bone-marrow-derived M2 macrophages prevented Dox-induced cardiac remodeling and injury. M2 macrophage transplantation transferred mitochondria to cardiomyocytes, alleviated doxorubicin-induced cellular stress, and reduced cardiac apoptosis [102].

Invariant natural killer T cells

Invariant natural killer T-cells (iNKT) are a subset of T-lymphocytes that express properties of both T cells and natural killer cells. Studies have shown that iNKT cells modulate cardiac tissue inflammation. For example, treatment with alpha-galactosylceramide (αGC), an activator of iNKT cells, prevented damage after MI. Another study looked at these cells in the context of Dox-induced cardiotoxicity and found that heart function was normal in mice treated with Dox and αGC simultaneously but not in mice treated with Dox only. Additionally, an analysis for fibrosis found that there was low collagen deposition in the mice treated with Dox+ αGC compared to mice with treated with Dox only. A qPCR analysis revealed that M2 macrophage expression was higher in mice treated with Dox+ αGC mice than in mice with Dox only. In this study, it was concluded that iNKT cell activation prevented Dox-induced LV dysfunction and cardiac fibrosis in mouse hearts hinting at a role for these cells in Dox-induced cardiotoxicity [103].

Conclusion

Significant advances have been made to delineate the role of inflammation and immune cells in cardiac injury and repair. In this review, we have highlighted our understanding of the role of innate immunity in cardiac diseases including anthracycline-induced cardiotoxicity. Danger signals released following an insult on cardiomyocytes are recognized by resident macrophages initiating inflammatory responses resulting in the recruitment of other immune cells, such as neutrophils, to remove the tissue debris. In tandem, activated cardiac fibroblasts and anti-inflammatory immune cells dampen inflammation and promote cardiac fibrosis. While cardiac fibrosis is important for cardiac repair, excessive tissue damage due to the continuous activation of the pro-inflammatory cascade can result in the upregulation of extra cellular matrices, which contribute to the development of late heart failure. To improve outcomes of Dox-induced cardiotoxicity and identify preventive interventions, it is important to understand the immune processes involved in mediating this cardiac injury and the repair processes that are subsequently induced (Figure 3). Included in this injury-reparative cycle are the roles of pattern-recognition receptors, macrophages (both M1 and M2), neutrophils, and iNK cells as well as inflammatory (TNFα, IL-6, IFN-γ) and immunosuppressive cytokines (IL-10, TGF-β) and the coordinated role of cardiac fibroblasts. Immune modulation by targeting neutrophils, macrophages, and cytokines to dampen inflammatory cascade and boost cardiac recovery and repair is highlighted as the new therapeutic approach to improve patient outcomes with cardiac disease.

One of the potential approaches is the liposomal formulation of Dox. The pegylated (polyethylene-glycol-coated) and non-pegylated liposomal formulation of Dox has been evaluated in clinical trials and they are currently in clinical use [104,105]. Liposomal Dox, in comparison to conventional Dox, has demonstrated significantly lower cardiotoxicity profiles with better cardiac safety but with well-preserved anti-tumor efficacy. The lower cardiotoxicity is associated with lower myocardial drug cardiotoxicity. Moreover, liposomal formulation prevents the direct contact of the cytotoxic agent with the vasculature, influencing the pharmacokinetics and biodistribution of the agent. This significantly reduces the accumulation in organs with dense endothelium such as the heart, while maintaining anti-tumor efficacy. Canine and porcine studies suggest the peak and overall drug concentration in the heart after liposomal Dox infusion was ~40% lower than conventional Dox [105]. This is associated with an increased expression of interferon-stimulated genes (ISGs) such as DHX58 and OAS1. This mediates pro-survival and DNA damage resistance response after Dox treatment. In marked contrast, conventional Dox is associated with higher cardiac concentration and fails to induce ISGs and subsequently drives cell death [105,106].

Despite extensive research, the molecular pathogenesis of Dox-induced cardiomyocyte injury is complex and incompletely understood. Importantly, these multi-factorial mechanisms are not independent and can occur simultaneously. While the majority of studies focus on apoptosis, recent studies suggest that Dox also induces inflammatory cell death such as necrosis, ferroptosis, and necroptosis. The identification of the key molecules that play critical roles in multiple pathways regulating Dox-induced cell death may represent an important therapeutic target. Pharmacological agents against the same target can potentially block multiple pathways simultaneously thus preventing Dox-induced cardiac cell damage. Furthermore, recent studies suggest innate immune cells, particularly neutrophils and M1 macrophages, amplify inflammatory cascade induced by Dox. Further studies are important to delineate the correlation between inflammatory mediators and the activation of immune cells. It is critically important to understand the crosstalk between immune cells and their role in Dox-induced cardiomyocyte death.

## Figures and Tables

**Figure 1 ijms-23-14649-f001:**
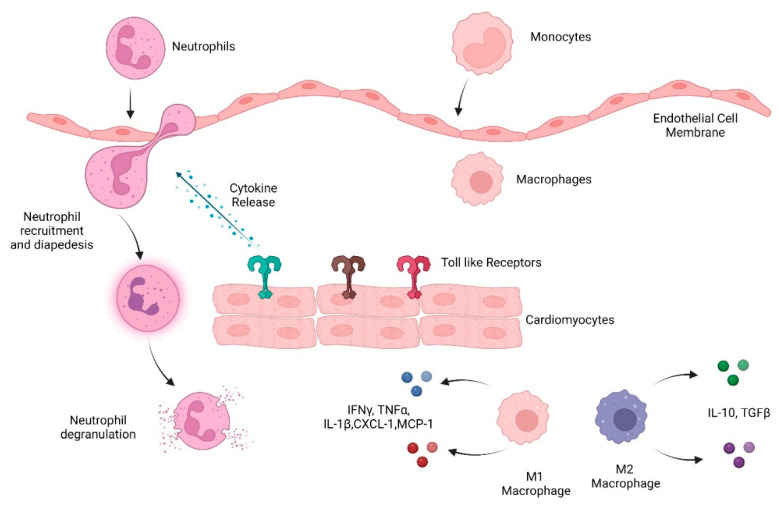
Overview of the innate immune system response to heart damage. Toll-like receptors detect damaged heart tissue and initiate an immune response including cytokine and chemokine release that recruits immune cells, including neutrophils, monocytes, and macrophages, to heart tissue.

**Figure 2 ijms-23-14649-f002:**
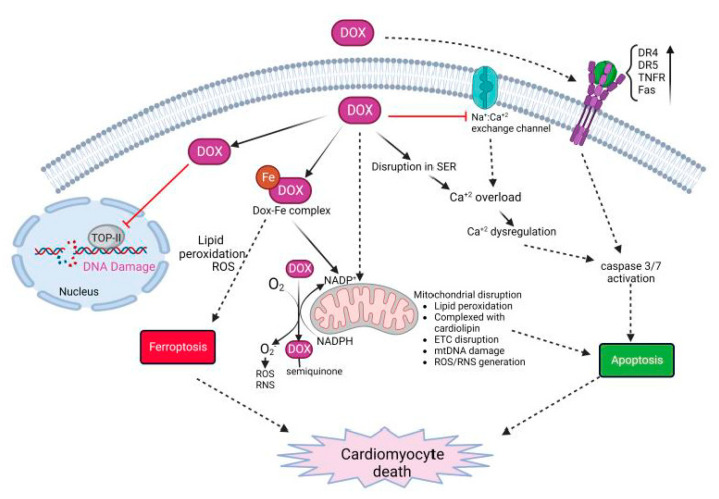
Molecular mechanisms on Dox-induced cardiomyocyte death. Dox treatment initiates multi-focal pathways including upregulation of death receptors, calcium overload, disruption in iron homeostasis, and generation of ROS/RNS. These events lead to oxidative stress, lipid peroxidation, DNA and protein damage, and cell death.

**Figure 3 ijms-23-14649-f003:**
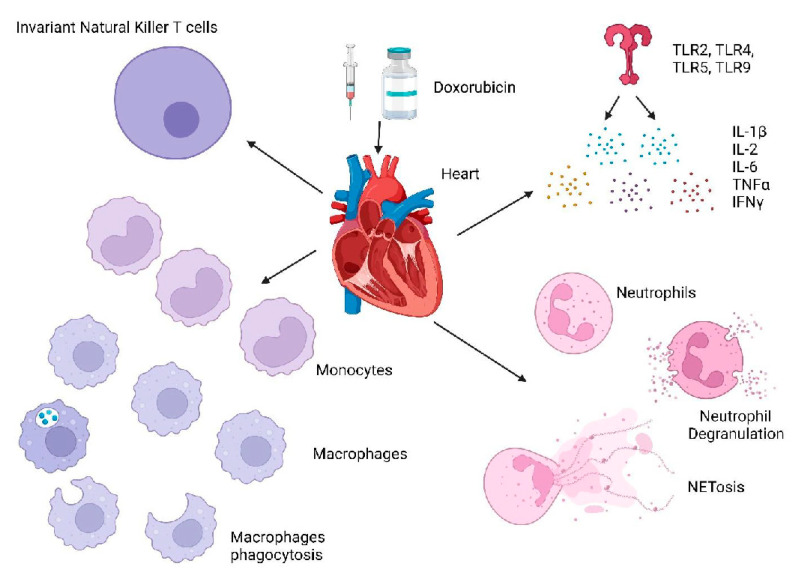
Overview of the Innate Immune System in Doxorubicin-Induced Cardiotoxicity: Multiple immune cells including monocytes, macrophages, neutrophils, and invariant natural killer T cells are shown to be involved in the Doxorubicin-induced cardiac damage in the heart.

## Data Availability

Not applicable.

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
