# Peer review of "The Innate Immune System in Cardiovascular Diseases and Its Role in Doxorubicin-Induced Cardiotoxicity"

_ijms, 2022, doi:10.3390/ijms232314649_

Round 1
Reviewer 1 Report
Dear aurhors,
The innate immunity involved in doxorubicin-induced cardiac injury is a urgent issues for the application of doxorubicin. The authors summarize the recent studies and provide a broad view about this issue. I suggest three points to improve the soundness of this study:
1. Please describe the recent knowledge about the doxorubicin-induced cardiomyocyte injury. The authors spend about a half of pages in describing the inflammatory mechanism appeared in heart. However, the initiation about doxorubicin-induced inflammation relies on cardiomyocyte injury. Therefore, the authors need to describe the recent mechanism of doxorubicin-induced cardiomyocyte injury so that the reader can get into the swing of things more quickly.
2. Please highlight the critical issues in delineating doxorubicin-induced inflammation. Do the authors think that the mechanism of doxorubicin-induced inflammation is completely resolved? Or there are some unsolved issues? Please describe them in the conclusion or future remarks.
3. Is applicable, please discuss the impact of novel formulation (e.g., liposome-coated doxorubicin) on doxorubicin-induced inflammation. New formulation of doxorubicin is known to affect the efficacy of tumor control and ADMET of doxorubicin. How about the impact of novel formulation in doxorubicin-induced inflammation?
Author Response
We would like to thank the reviewers for their valuable insight and suggestions, please see our responses below.
- Please describe the recent knowledge about the doxorubicin-induced cardiomyocyte injury. The authors spend about a half of pages in describing the inflammatory mechanism appeared in heart. However, the initiation about doxorubicin-induced inflammation relies on cardiomyocyte injury. Therefore, the authors need to describe the recent mechanism of doxorubicin-induced cardiomyocyte injury so that the reader can get into the swing of things more quickly.
Response: We would like to thank the reviewers for their suggestion. We have included a new section titled Section 3. Dox-induced cardiomyocyte injury where we have described recent mechanism of dox-induced cardiomyocyte injury on Page 9 as noted by the tracked changes.
- Please highlight the critical issues in delineating doxorubicin-induced inflammation. Do the authors think that the mechanism of doxorubicin-induced inflammation is completely resolved? Or there are some unsolved issues? Please describe them in the conclusion or future remarks.
Response: We would like to thank the reviewers for this valuable insight. We have included a further discussion of critical issues in delineating Dox-induced inflammation in the Conclusion portion of the review on Page 17 as noted by the tracked changes.
- Is applicable, please discuss the impact of novel formulation (e.g., liposome-coated doxorubicin) on doxorubicin-induced inflammation. New formulation of doxorubicin is known to affect the efficacy of tumor control and ADMET of doxorubicin. How about the impact of novel formulation in doxorubicin-induced inflammation?
Response: We would like to thank the reviewers for this suggestion. We have described the impact of novel formulations on Dox-induced inflammation in the Conclusion section on Page 17 as noted by the tracked changes.
Reviewer 2 Report
The manuscript extensively describes the molecular mechanisms of cardiotoxicity.
I have minor comments. Also, I suggest authors to explain all the abbreviations.
I will make some comments as follows:
1. What is the main question addressed by the research?
The research is a review of principal actors (immune cells, inflammation cytokines and chemokines) influencing cardiac injuries and cardiotoxicity of Doxorubicine.2. Do you consider the topic original or relevant in the field? Does it
address a specific gap in the field? The topic addresses a specific issue, cardiac toxicity, which gains more and more importance in the context of increased survival rate in some cancers
3. What does it add to the subject area compared with other published
material? The manuscript is a comprehensive review of the topic.
4. What specific improvements should the authors consider regarding the
methodology? What further controls should be considered? The manuscript does not have an original interpretation of the data, yet is useful to have all the information in one place.
5. Are the conclusions consistent with the evidence and arguments presented
and do they address the main question posed? YES
6. Are the references appropriate? YES
7. Please include any additional comments on the tables and figures. The manuscript has one single figure, which is of good quality and relevant for the text.

Author Response
We would like to thank the reviewers for all their comments, see our responses below
- I have minor comments. Also, I suggest authors to explain all the abbreviations.
Response: We would like to thank the reviewers for this suggestion, we have explained all the abbreviations and included an appendix of terms before the References.
2. We have addressed all the minor comments as indicated in the uploaded copy.
Reviewer 3 Report
In this review, the author summarized the innate immune system in cardiovascular diseases and doxorubicin-induced cardiotoxicity.
Neutrophiles and macrophages were discussed in detail. It is a well-structured and clearly represented review paper. Recent studies of the top including the study from the author’s lab was included and discussed in the review paper.
I only have a few minor comments here.
1. In section 2.3, Neutrophils in CVDs: It is not clear in some
sentences if the neutrophils discussed are the neutrophils in the
peripheral blood or those infiltrate into the cardiac tissue. Please
to be specific.
2. In section 3.3, macrophages:
In a study conducted to examine the role of NLRP3, …. A comma was missing here.
3. To be consistent with the prior discussion, in the section 3.3, I
would suggest introducing neutrophiles first and then macrophage.
4. Figure1 is not clear to me. It is not clear of what does iNKT do in
doxorubicin-induced cardiotoxicity. Also, it is not clear whether
they are the cell subsets in the tissue or in the peripheral blood.
The author could try to make clear either in the figure or in the
legend.
Author Response
We would like to thank the reviewers for their comments and valuable insight. Here are our responses below:
1.In section 2.3, Neutrophils in CVDs: It is not clear in some sentences if the neutrophils discussed are the neutrophils in the peripheral blood or those infiltrate into the cardiac tissue. Please to be specific.
Response: We would like to thank the reviewers for this suggestion, we have added clarification in the neutrophils in CVD section as noted by the tracked changes.
2.In section 3.3, macrophages: In a study conducted to examine the role of NLRP3, …. A comma was missing here.
Response: We would like to thank the reviewers for this valuable comment, we have added the comma.
3. To be consistent with the prior discussion, in the section 3.3, I would suggest introducing neutrophils first and then macrophage.
Response: We would like to thank the reviewers for their suggestion. We have taken their advice and switched the neutrophils section to be first as noted by the tracked changes.
4.Figure1 is not clear to me. It is not clear of what does iNKT do in doxorubicin-induced cardiotoxicity. Also, it is not clear whether they are the cell subsets in the tissue or in the peripheral blood. The author could try to make clear either in the figure or in the legend.
Response: We would like to thank the reviewer for this insight. We have clarified the role of iNKT further in the section of Invariant natural killer T cells as noted by the tracked changes on page 17.
Reviewer 4 Report
1. The title can be changed to “The Innate Immune System in Cardiovascular Diseases and its role in Doxorubicin-Induced Cardiotoxicity”.
2. Only one figure for a comprehensive review is not good enough. I suggest the authors add more figures to conclude the mechanisms of the immune cells' response to DOX-induced cardiotoxicity.
Author Response
We would like to thank the reviewers for their comments and suggestions, please see our responses below:
- The title can be changed to “The Innate Immune System in Cardiovascular Diseases and its role in Doxorubicin-Induced Cardiotoxicity”.
Response: We would like to thank the reviewer for this suggestion, we have changed the title accordingly as noted by the tracked changes.
- Only one figure for a comprehensive review is not good enough. I suggest the authors add more figures to conclude the mechanisms of the immune cells' response to DOX-induced cardiotoxicity.
Response: We would like to thank the reviewers for this valuable insight. We have added 2 new figures to the review to explain the mechanism better as noted by the tracked changes in Figure 1 and Figure 2.
Round 2
Reviewer 1 Report
Dear authors,
Thank you for your kind response to my suggestion. I have no suggestion to your review.
Reviewer 4 Report
No further comments.